# Smoking cessation and prognosis during long-term follow-up after stroke, TIA, and acute coronary syndrome—results from the randomized controlled NAILED trial

**Anna-Lotta Irewall** *, **Lina Åslund, Joachim Ögren, Thomas Mooe**

Department of Public Health and Clinical Medicine, Umeå University, Östersund, Sweden

* anna-lotta.irewall@umu.se

## Abstract

**Data Availability Statement:** As open access to individual-level data was not specified in the

### Background and aims

About 50% of patients continue to smoke after stroke and myocardial infarction. We aimed to assess the effect of a multiple risk factor intervention on long-term smoking cessation and to explore a possible association between early smoking cessation and long-term prognosis.

### Material and methods

Consecutive patients with stroke/TIA/acute coronary syndrome (ACS) at Östersund Hospital during 2010–2014 were included, randomized to intervention or usual care (1:1), and followed through 2017. This substudy included participants that reported current smoking during hospitalization and were alive at 1 month post discharge when the intervention began (n = 321). The smoking cessation intervention was part of a telephone-based, multiple risk factor intervention delivered by a nurse and consisted of brief advice delivered annually. Smoking cessation at the last follow-up was analyzed as the primary outcome. Smoking cessation at other time points and association between early smoking cessation and prognosis (CV events, survival) were secondary outcomes.

### Result

After a mean follow-up of 4.2 years, 171 participants reported nonsmoking, with no significant difference between the intervention and control group (50.3% vs. 56.3%, absolute difference 5.9%, 95% CI −5.0 to 16.7, p = 0.286). Of these, 80.7% had stopped smoking within 1 month after discharge. The intervention did not improve smoking cessation proportions in the long or short term, and there was no apparent effect on smoking cessation attempts or sustained abstinence. Smoking cessation within 1 month was associated with lower all-cause mortality (HR 0.52, 95% CI 0.32–0.87), and there was a nonsignificant trend towards a lower incidence of CV events (HR 0.71, 95% CI 0.45–1.12).

original application approved by the ethics committee, the underlying data is only available upon reasonable request. Please contact the corresponding author or registrator@umu.se, Department of Public Health and Clinical Medicine, Umeå University.

**Funding:** TM received funding from the Unit of Research, Development and Education, Region Jämtland Härjedalen (JLL-376981, JLL-377161), and the Swedish Heart–Lung Foundation (20140541). The funders had no role in the study design, data collection and analysis, decision to publish, or preparation of the manuscript. Unit of Research, Development and Education, Region Jämtland Härjedalen: https://www.regionjh.se/forskningutveckling.4.15591b8415700f7566b3b361.html Swedish Heart–Lung Foundation: https://www.hjart-lungfonden.se/?gad_source=1&gclid=

## Conclusion

Annual brief advice by a nurse as part of a multiple risk factor follow-up did not improve long-term smoking cessation after stroke/TIA/ACS. Continued smoking past 1 month was associated with worse prognosis.

## Trial registration

ISRCTN registry ISRCTN96595458, ISRCTN23868518, ISRCTN30433343.

## Introduction

Tobacco smoking is one of the most important risk factors for cardiovascular disease [1–3]. In patients with established cardiovascular disease, smoking cessation has been associated with improved survival [4,5] and reduced numbers of new cardiovascular events [5,6], with improvement seen already within the first years following smoking cessation [4,7]. Abstinence from smoking is therefore an important goal for secondary prevention, and offering patients support to help them quit is a high priority recommendation in national [8,9] and international guidelines [10–12].

In Sweden, around 50% of patients stop smoking after stroke or myocardial infarction, with no clear improvement of this proportion seen during the last decade [13–15]. Similarly, results from the cross-sectional EUROASPIRE surveys have found no improvement in the smoking cessation rate after coronary events during the 21st century [16,17]. This lack of improvement indicates a need for improved strategies and/or implementation to support smoking cessation in clinical practice. Several reports have also indicated a need for improved follow-up when it comes to other modifiable risk factors in patients with manifest cardiovascular disease [16,18–20]. Smoking cessation intervention as part of a multiple risk factor follow-up program may, therefore, be the most effective way to reduce recurrence of cardiovascular events in this high-risk group.

Previous studies of smoking cessation interventions indicate that counseling provided at multiple occasions during more than 1 month may increase the proportion of patients with cardiovascular disease that quit smoking [21–24], whereas less extensive counseling interventions, initiated in-hospital or in other settings and with less than 1 month of follow-up, seem to be insufficient [21,24]. Previous studies of multiple risk factor interventions indicate that high-intensity interventions, commonly with multiple contacts during the first months after a cardiovascular event [21,22] or sometimes longer [23,25], may increase the smoking cessation rate, but the design and results of available studies are heterogeneous [21–23,25–29]. In addition, common characteristics of successful interventions are difficult to distinguish because the smoking cessation component of these interventions has not always been well described [21,22,25,26,30–32], study samples have often been small [21,22,25,30,33], and there are very few examples of individual studies with significant results [21–23,25]. Few studies have evaluated less resource-demanding multiple risk factor interventions [33,34] or programs with infrequent contacts during long-term follow-up [34]. Thus, it is difficult to draw any firm conclusions regarding the effect of such interventions on smoking cessation.

The Nurse-based Age-independent Intervention to Limit Evolution of Disease (NAILED) trial was a pragmatic, secondary preventive, randomized controlled trial with a population-based recruitment approach, performed at a single center in Sweden during the years 2010–

2017. The aim of the trial was to investigate if long-term, nurse-based telephone follow-up could improve secondary preventive outcome after stroke/transient ischemic (TIA) attack and acute coronary syndrome (ACS). Previously published results from the NAILED trial have shown that this relatively simple but systematic form of risk factor intervention achieved significantly lower blood pressure (BP) and low-density-lipoprotein cholesterol (LDL-C) levels as well as a reduced recurrence of major cardiovascular events compared with usual care during long-term follow-up [35–37]. The aims of the current substudy were (1) to assess if the NAILED intervention also increased the smoking cessation rate compared with usual care, and (2) to explore the possible association between smoking cessation and prognosis.

## Material and methods

### Study design

The current study is a substudy of the NAILED trial, which was an open, randomized, controlled, long-term trial with two parallel groups (1:1).

### Participants

The NAILED trial included consecutive patients admitted to Östersund Hospital, Jämtland County, Sweden due to acute stroke/TIA (Jan 1, 2010 to Dec 31, 2013) or acute coronary syndrome, ACS (Jan 1, 2010 to Dec 31, 2014). ACS was defined as acute myocardial infarction (AMI) or unstable angina (UA). Patients were excluded if they were considered unable to participate in telephone-based follow-up due to severe aphasia, deafness, or dementia. Patients without indication for intensive secondary prevention due to severe, often terminal illness, and patients participating in other ongoing trials were also excluded. The current study included NAILED participants who were current smokers at the time of their index event and who remained in the trial at the first follow-up 1 month after discharge. Current smoking was defined as self-reported, habitual use of smoked tobacco of any duration and quantity but including at least the preceding 6 months—that is, patients who made a decision to stop smoking in-hospital or who reported smoking cessation during the preceding 6 months were still counted as current smokers according to the definition.

### Randomization

Patients that agreed to participate in the study were randomly allocated to the intervention group or the control group (1:1). The randomization was computer generated (by TM) in blocks of four and stratified for sex and, depending on the qualifying event, either degree of disability according to the modified Rankin Scale (stroke/TIA; modified Rankin scale 0–2 vs. 3–5) or the type of qualifying event (ACS: AMI vs. UA). The randomised allocation sequences was put into sequentially numbered, opaque, sealed envelopes by TM and the process of enrolment and random group assignment was performed by study nurses. The group assignment was not blinded to the participants, the study team or other care givers.

### Intervention

The intervention consisted of a telephone-based, nurse-led, annual follow-up of modifiable risk factors including blood pressure, blood lipids, diet, physical activity, and smoking habits. The average duration of a phone call was about 10–15 minutes but was individualized for each participant. The time spent on lifestyle counseling was not registered and thus cannot be described in further detail. For participants with BP or LDL-C above the target level, the pharmacological treatment was adjusted and the participant was followed every 4 weeks until the

target level was met or no further improvement was considered possible. Patients were asked about their smoking habits at every yearly follow-up, but not at the extra follow-ups due to medication adjustments. If a participant was smoking, they received brief information about smoking as a risk factor and were encouraged to stop. They also received information regarding how primary care could help them stop smoking. Pharmacotherapy for smoking cessation was not offered as part of the intervention, but could be prescribed to participants in both groups through primary health care.

## Control

Control group participants received secondary preventive follow-up according to usual care. For most patients with an ACS, that included follow-up at the cardiology outpatient clinic within 4 weeks (nurse) and 3 months (physician) after hospital discharge (11). Thereafter most patients were followed by a general practitioner at their primary health care center. The study nurse also contacted all participants annually to collect study data including risk factor levels, smoking status, and current medication.

## Outcomes

The primary outcome was the proportion of participants who reported nonsmoking (defined as self-reported nonsmoking) in the intervention group compared with the control group at the last follow-up. Smoking status was a predefined outcome of the NAILED trial as specified in the previously published study protocol [38,39], but assessment at the last follow-up as the primary outcome was a revision to the original protocol. The revision was made because the long-term result was considered more clinically relevant and also a better outcome to reflect the long-term intervention. Between-group differences in smoking status at 1 year, 3 years, and at any time point during follow-up were analyzed as secondary outcomes. Secondary outcomes also included exploratory analysis of the association between smoking cessation and (1) population characteristics, (2) recurrence of cardiovascular events, and (3) all-cause death. Cardiovascular events comprised nonfatal stroke, nonfatal AMI, cardiac revascularization, and cardiovascular death. The events are defined in further detail in a previous publication [37] and as a (S1 File).

## Data collection

Baseline characteristics including smoking status of the study population were collected by study nurses during the hospitalization for the qualifying event. The data were collected through patient interviews and review of the medical records. Smoking status was reassessed at 1 month and thereafter annually through the conclusion of the trial by Dec 31, 2017. Depending on when the patient was included in the trial, the maximal duration of follow-up varied from 3 to 8 years.

To identify recurrent cardiovascular events during the study period, all discharge records for hospitalization at the Department of Internal Medicine during the study period were reviewed. In addition, the hospital inpatient register was used to search for relevant discharge diagnoses to identify events occurring at other hospital departments, and the Swedish Coronary Angiography and Angioplasty Register (SCAAR) was used to identify events of cardiac revascularization. Identified events were evaluated strictly according to predefined criteria. The review process was blinded to group allocation and performed by four medical doctors who were part of the study team. Data regarding cardiovascular events and mortality were also collected until the conclusion of the trial for participants that had discontinued the study annual follow-ups at an earlier time point. Participants that moved out of the county/country

were followed until the date of migration/last registered healthcare contact because documentation of any event occurring thereafter would not be available for the study team.

## Statistical method

Population characteristics were described as percentages for categorical variables and mean values with standard deviation for continuous variables. Differences in population characteristics based on (1) the allocated treatment group (intervention vs. control), and (2) smoking status (yes/no) were compared using chi-square or t-test as appropriate. Chi-square was also used to compare smoking status between allocated groups at the different time points specified in the outcome section above. Association between early (within 1 month) smoking cessation and (1) new cardiovascular events, and (2) total mortality was analyzed using Kaplan-Meier (KM) survival analysis with log-rank testing for group comparison. Participants contributed time at risk from discharge until the first occurrence of an event, death due to any cause, emigration, or Dec 31, 2017, whichever occurred first. A Cox proportional-hazards model was used to calculate the crude hazard ratio for CV events/total mortality. As a sensitivity analysis, we also performed the Cox proportional-hazards analysis using smoking status as a time-varying variable, taking into account smoking status beyond 1 month. In addition, the KM analysis of new cardiovascular events was also complemented by a competing risk analysis according to Fine-Gray, accounting for all-cause mortality as a potential competing risk variable. All analysis were performed according to the intention-to-treat principle. Participants that withdrew from the study were treated conservatively by being included in the denominator as continued smokers. Statistical analysis was performed with SPSS software, version 28.0 (IBM Corp. Amonk, New York, USA) and SAS Ver. 9.4 (SAS Institute Inc, Cary, North Carolina, USA).

## Trial registration

The NAILED risk factor control trial, including smoking as a prespecified outcome, is registered in the ISRCTN registry with separate registrations for the ACS (ISRCTN96595458) and stroke/TIA cohort (ISRCTN23868518). In addition, the NAILED cardiovascular outcome trial, including cardiovascular outcome for both cohorts, is registered as a separate trial (ISRCTN30433343). The strict ICMJE requirement of prospective registration of clinical trials came to our attention when recruitment had already begun. Therefore, all the trials included in the NAILED project were retrospectively registered. The authors confirm that all ongoing and related trials for this intervention are registered.

## Ethics

This study was a part of the NAILED trial, which previously received ethical approval from the Regional Ethics Committee in Umeå (Dnr 09-142M). Smoking status was a predefined outcome in the original study protocol. The patients included in the study have signed an informed, written consent document. The study protocol, as approved by the ethics committee, is available as a (S2 File).

## Results

### Baseline characteristics

Out of 3228 patients with acute stroke/TIA/ACS assessed for eligibility in the NAILED trial, 1890 were included. Among those included, 332 (17.6%) reported current smoking at baseline. The corresponding proportion was similar (19.3%) among eligible patients that declined participation (n = 384). Among included smokers, 1 participant died and 10 participants

withdrew during the first month after discharge, before the intervention began. Those that remained in the study at 1 month (n = 321) constitute the study population of the current study (Fig 1).

Baseline characteristics are described in Table 1. The mean age was 63.8 years (SD 10), 33.3% (n = 107) were women, and 20.6% (n = 66) had had previous vascular events (AMI/cardiac revascularization/stroke/TIA/peripheral artery disease) prior to the index event. The types of index events were distributed as follows: 85 participants with stroke (26.5%), 41 participants with TIA (12.8%), and 195 participants with ACS (60.7%). No significant differences were found in baseline characteristics between the intervention group (n = 161) and the control group (n = 160), but there was a numerical trend toward more participants with previous vascular events in the control group (p = 0.092).

## Follow-up time

The mean follow-up time was 4.2 years, but varied from 1 month to 8 years. There was no significant difference in follow-up length between the two groups (p = 0.446).

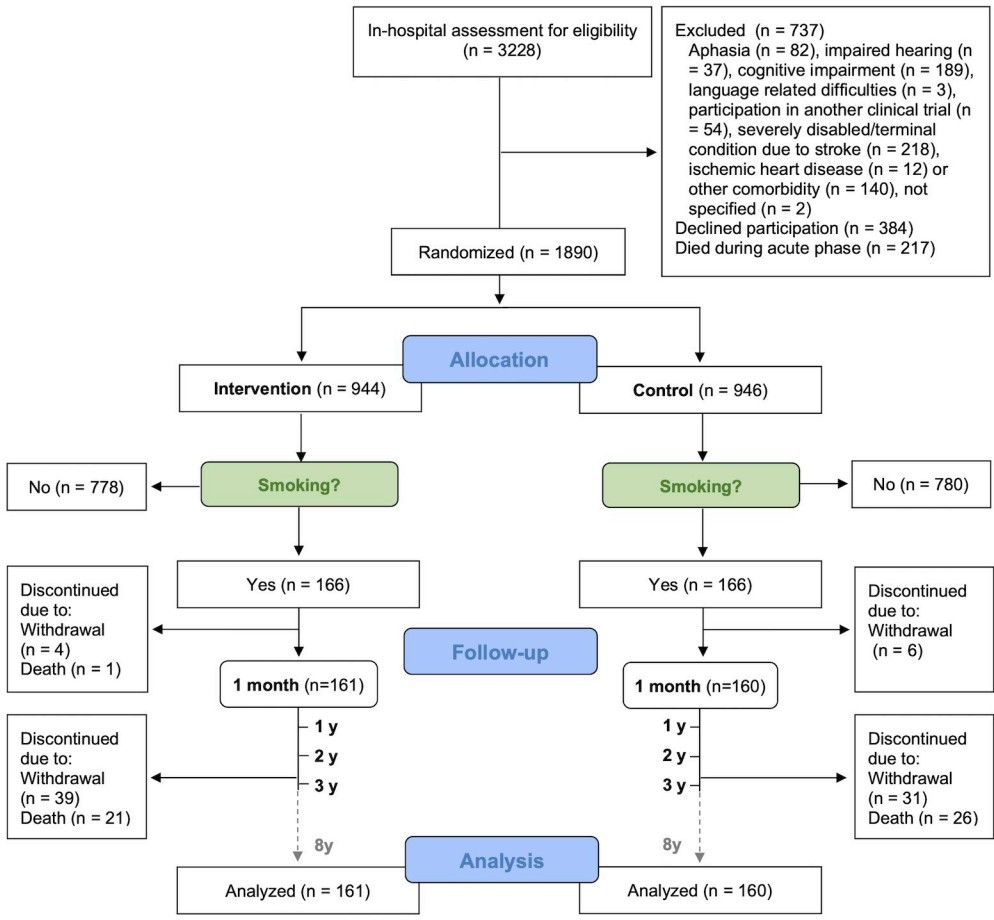

**Fig 1. Study flowchart.** Participants were assessed for eligibility, included, and randomized in hospital, whereas the intervention started 1 month after discharge. Smokers that remained in the study at 1 month constitute the study population of the current substudy. The maximum duration of intervention/usual care follow-up was 3 to 8 years, depending on the year of inclusion. All analyses were performed according to the intention-to-treat principle. Smoking status at the last follow-up occasion for each participant was used to analyze the primary outcome. Participants were followed for cardiovascular and mortality outcomes until Dec 31, 2017, irrespective of withdrawal from the study's annual follow-ups at an earlier time point.

### Smoking cessation

In total, 171 participants (53.3%) reported nonsmoking at their last follow-up: 81 patients (50.3%) in the intervention group and 90 patients (56.3%) in the control group, with no significant difference between groups (absolute difference 5.9%, 95% CI −5.0 to 16.7, p = 0.286). Correspondingly, we found no between-group difference in smoking status at 1 year or in the proportions that reported smoking cessation at some time point during the course of the study (Fig 2). However, at 3 years there was a significant difference in favor of the control group (absolute difference 12.8%, 95% CI 1.3–23.9, p = 0.029).

About one-third (n = 101, 31.5%) of the participants reported continued smoking at all their follow-ups. Among those who reported nonsmoking at their last follow-up, 80.7% (n = 138) had already reported nonsmoking at 1 month. Among patients that reported nonsmoking at 1 month (n = 179, 55.8%), 33.5% (n = 60) had at least one relapse during follow-up, and 77.1% (n = 138) reported nonsmoking at their last follow-up, again with no significant difference between groups.

### Population characteristics associated with smoking cessation

Participants that reported nonsmoking at their last follow-up were slightly younger (62.8 vs. 64.9 years, p = 0.050) and a higher proportion had had ACS (66.1% vs. 54.7%, p = 0.037) as their index event compared with those that reported continued smoking (S1 Table). Characteristics stratified for smoking cessation at 1 month showed a similar pattern, with the addition of a lower proportion of patients with TIA as the index event (5.6% vs. 21.8%, p ≤ 0.001), a trend towards lower prevalence of previous manifestations of cardiovascular disease (16.8 vs. 25.4%, p = 0.059), a higher proportion with antihypertensive treatment (88.3 vs. 79.6%, p = 0.033), and, as expected considering the distribution of index events, a higher proportion with dual antiplatelet therapy (52.1 vs. 63.7, p = 0.037) at discharge in the nonsmoking group.

### Smoking cessation and recurrence of cardiovascular events and total mortality

A total of 74 participants (23.0%) reached the composite endpoint of CV events and 64 participants (19.9%) died from any cause (Table 2). The incidences of CV events and all-cause death were 4.9 and 3.8 per 100 years at risk, respectively. Smoking cessation at 1 month was associated with a lower cumulative incidence of all-cause death (Fig 3B). A similar trend was seen for CV events, but the difference was not statistically significant (Fig 3A). The sensitivity analyses, including smoking status as a time-varying variable and accounting for all-cause mortality as a potential competing risk variable, did not change any results.

## Discussion

In this study, we found no effect of brief smoking cessation advice delivered by a nurse as part of telephone-based follow-up of cardiovascular risk factors in patients with stroke, TIA, or acute coronary syndrome. Compared with usual care, the intervention did not improve smoking cessation proportions in the long or short term, it did not seem to increase smoking cessation attempts overall, and we found no effect regarding sustained abstinence among those that quit smoking shortly after the index event (i.e., before the intervention began). Smoking cessation at 1 month was associated with improved survival during long-term follow-up. A similar trend was seen for CV events, but this difference was not statistically significant. Besides a positive effect of smoking cessation itself, differences in baseline characteristics between those that

**Table 1. Baseline characteristics.**

|  | Intervention | Control |
|---|---|---|
| N | 161 | 160 |
| Women, N (%) | 56 (34.8) | 51 (31.9) |
| Age, years, mean (SD) | 62.8 (10.2) | 64.8 (9.7) |
| Low education level, N (%)* | 82 (51.3) | 84 (52.8) |
| BMI, median (iqr) | 27.7 (5.3) | 27.0 (4.7) |
| eGFR, mL/min, median (iqr)† | 87.6 (17.8) | 86.2 (16.7) |
| Index event: ACS, N (%) | 100 (62.1) | 95 (59.4) |
| *STEMI* | 43 | 34 |
| *NSTEMI* | 54 | 54 |
| *UA* | 3 | 7 |
| Index event: stroke, N (%) | 38 (23.6) | 47 (29.4) |
| *Ischemic* | 36 | 44 |
| *Hemorrhage* | 2 | 3 |
| Index event: TIA, N (%) | 23 (14.3) | 18 (11.3) |
| mRS 0–2, N (%)‡ | 51 (83.6) | 58 (89.2) |
| Previous vascular events, N (%) | 27 (16.8) | 39 (24.4) |
| Previous ischemic heart disease, N (%) | 19 (11.8) | 28 (17.5) |
| Previous stroke, N (%) | 7 (4.3) | 13 (8.1) |
| Previous TIA, N (%) | 4 (2.5) | 4 (2.5) |
| Peripheral artery disease, N (%) | 3 (1.9) | 5 (3.1) |
| Congestive heart failure, N (%) | 2 (1.2) | 1 (0.6) |
| Atrial fibrillation, N (%) | 12 (7.5) | 19 (11.9) |
| Diabetes, N (%) | 20 (12.4) | 21 (13.1) |
| CKD (eGFR<60 mL/min), N (%)† | 13 (8.1) | 14 (8.9) |
| Hypertension, N (%) | 74 (46.0) | 77 (48.1) |
| Antihypertensive drug (s), N (%) | 138 (85.7) | 133 (83.1) |
| *1 drug* | 33 | 33 |
| *2 drugs* | 69 | 53 |
| *≥3 drugs* | 36 | 47 |
| Lipid lowering drug, N (%) | 146 (90.7) | 145 (90.6) |
| Antiplatelet drug, N (%) | 151 (93.8) | 152 (95.0) |
| *DAPT* | 91 (56.5) | 97 (60.6) |
| Warfarin, N (%) | 11 (6.8) | 9 (5.6) |

There were no significant differences in baseline characteristics, but there was a numerical trend toward a higher proportion of previous vascular events in the control group (p = 0.092). Low education level was defined as no more than 10 years of formal education. Previous ischemic heart disease includes previous acute myocardial infarction, percutaneous coronary intervention or coronary artery by-pass grafting. Previous vascular events was a composite of previous ischemic heart disease, previous stroke or TIA, or previous peripheral artery disease. Drug treatment variables refers to treatment at discharge. SD, standard deviation; ACS, acute coronary syndrome; STEMI, ST elevation myocardial infarction; NSTEMI, non–ST elevation myocardial infarction; UA, unstable angina; BMI, body mass index; eGFR, estimated glomerular filtration rate; CKD, chronic kidney dysfunction; TIA, transient ischemic attack; DAPT, dual antiplatelet therapy.

* Missing value for 1 intervention group participant and 1 control group participant.

† Missing value for 1 intervention group participant and 2 control group participants.

‡ Participants with stroke/TIA as the qualifying event.

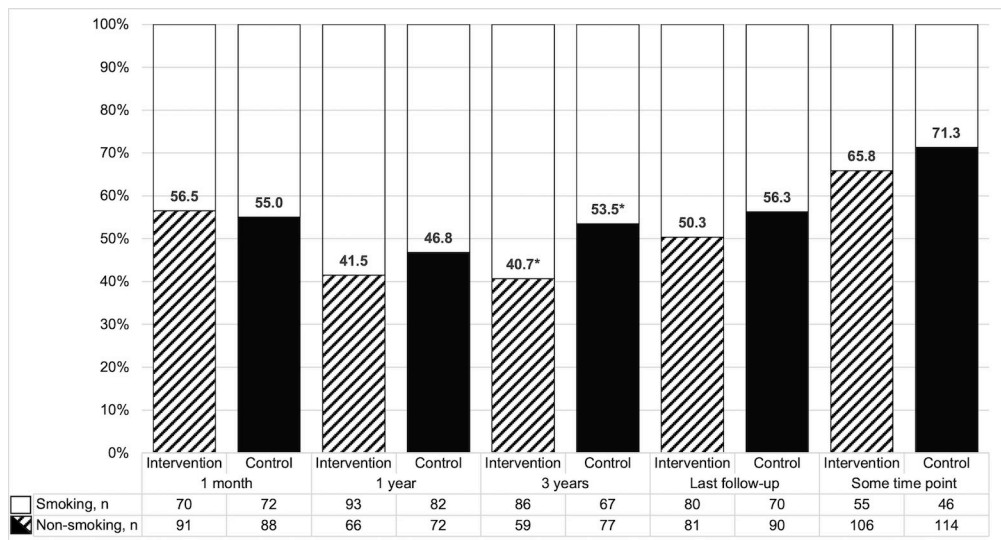

**Fig 2. Self-reported smoking cessation.** Proportions according to intention-to-treat: The denominator includes all patients alive at the respective time points and continued smoking was assumed for participants with missing values due to study withdrawal or other reasons. The numbers of cases with missing values in the intervention and control groups, respectively, were 0 vs. 1 at 1 year, and 21 vs. 9 at 3 years. *Significant between-group difference in favor of the control group (p = 0.029).

quit smoking early after the index event and those that did not may have contributed to the observed difference in prognosis.

Smoking cessation support is a high priority recommendation of current guidelines [9–12], but recommendations regarding specific strategies are often vague [8,11,12] and sometimes include brief counseling [9,10,12], despite little evidence for the effectiveness of such interventions among patients with manifest cardiovascular disease. In this context, our study is an important contribution and the results are in line with previous studies pointing at no short-term [31,33,34] or long-term [34] effect of counseling delivered on a single occasion [31,33] or repeatedly [34] during long-term follow-up. In a previous Swedish single-center RCT, lifestyle counseling and general risk factor assessment by a nurse at 3 months after stroke did not reduce the proportion of current smokers at 12 months compared with usual care [33]. In another RCT, performed among patients with stroke or high-risk TIA in Austria, multidisciplinary follow-up at 3 months and access to a web-based platform with educational and

**Table 2. Association of self-reported smoking cessation at 1 month with incidence of new cardiovascular events and all-cause death.**

|  | Smoking, N (%) | Non-smoking, N (%) | Absolute difference, % (95% CI) | HR (95% CI) |
|---|---|---|---|---|
| **Cardiovascular events**[*] | 37 (26.1) | 37 (20.7) | 5.4 (−3.9 to 14.7) | 0.71 (0.45–1.12) |
| Cardiovascular death | 7 | 5 |  |  |
| AMI (nonfatal) | 7 | 10 |  |  |
| Stroke (nonfatal) | 12 | 8 |  |  |
| Coronary revascularization | 11 | 14 |  |  |
| **All-cause death** | 36 (25.4) | 26 (14.5) | 10.8 (2.0–19.6) | 0.52 (0.32–0.87) |

AMI, acute myocardial infarction; STEMI, ST elevation myocardial infarction; NSTEMI, non–ST elevation myocardial infarction; PCI, percutaneous coronary intervention; CABG, coronary artery bypass grafting; TIA, transient ischemic attack.

[*]The first event to occur was counted.

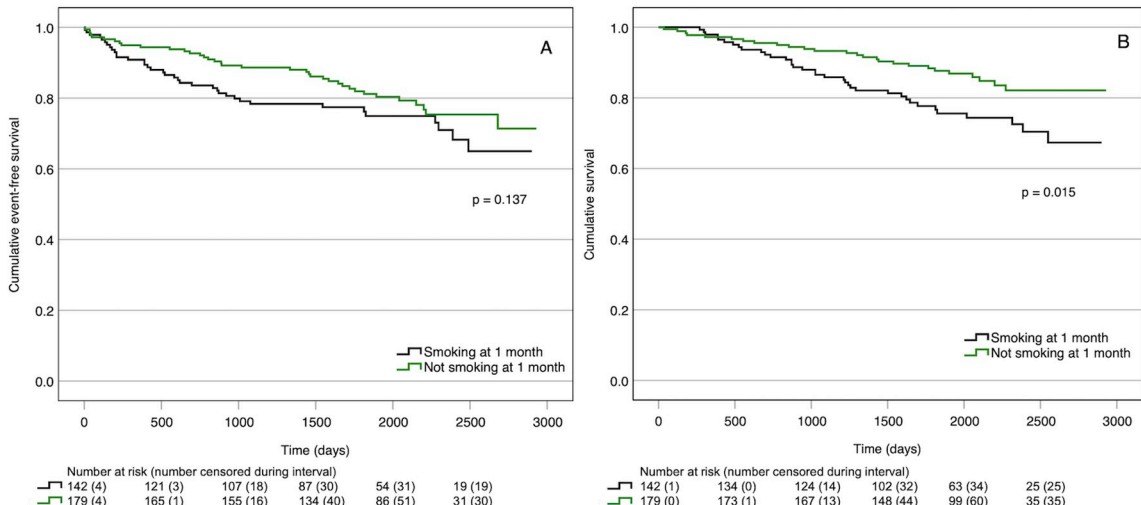

**Fig 3.** Association of smoking cessation at 1 month with occurrence of (A) new cardiovascular events and (B) all-cause death.

interactive elements had no effect on smoking cessation rates at 12 months [31]. In a single-center RCT from Norway, no effect on smoking cessation rates at any time point was found during 5 years of follow-up [34]. This Norwegian study compared a nurse-based, long-term, secondary preventive follow-up program with usual care among consecutive patients aged <80 years and hospitalized due to MI or scheduled PCI/CABG. The intervention had slightly more extensive follow-up during the first year compared with NAILED and included an offer of pharmacotherapy (nicotine replacement therapy/varenicline) as part of the intervention.

The role of smoking as a causal factor in cardiovascular disease is well established [40]. It has also been shown in several observational studies that smoking cessation is associated with reduced recurrence of cardiovascular events and death [5,6,41]. The relative risk reduction for all-cause death in our study was almost 50%, which is similar [41] or perhaps even more pronounced [5,6] than the estimates seen in previous observational studies. It should, however, be noted that the confidence interval in our study was wide and the true risk reduction could therefore be considerably larger or smaller. Similarly, the low number of participants contributing time at risk at the end of follow-up introduces uncertainty regarding the results for the cardiovascular event outcome, possibly contributing to a type 2 error. Also, as in previous studies, it is difficult to isolate the effect of smoking cessation alone, since smoking cessation was also associated with other characteristics that were of prognostic importance such as age, the type of index event, and (at a borderline significance level) previous vascular events. Age is inevitably associated with death. Previous manifestations of vascular events could indicate more severe vascular disease and thereby a worse prognosis. It can also be speculated that patients who continue to smoke despite having repeated manifestations of vascular disease are a rather select group that could have additional prognostic characteristics that are perhaps difficult to measure, such as other lifestyle-related factors, social context, or background. Few RCTs of smoking cessation interventions have evaluated their effects on cardiovascular and mortality outcomes. Mohiuddin et al. compared the addition of an intensive, postdischarge smoking cessation program based on multiple group sessions to in-hospital counseling alone in patients with ACS or decompensated congestive heart failure. They found an effect on mortality with a point estimate (RRR 77%; 95% CI 27%–93%; p = 0.014) suggesting a larger effect than in the observational studies. This was, however, a small study with few events, which limits the precision of the point estimate, and the follow-up was only 24 months [4].

Having a potentially life-threatening event such as a myocardial infarction or a stroke may be an eye-opener, increasing a smoker's motivation to quit. Indeed, high smoking cessation rates in the early time period following a myocardial infarction or stroke has been commonly observed in many previous studies [23,28,30,31,34,42,43] and, of note, these proportions are generally larger than any additional effect achieved by smoking cessation interventions [21,22,24,44,45]. In line with previous studies [23,28,30,31,34,42,43], we found that most patients that quit smoking after a cardiovascular event did so shortly after the event. Also, the rates of additional quitting attempts and relapse during the first year were very similar to previously observed rates [34,43]. In our study, smoking cessation, especially early smoking cessation, was more common among patients with ACS as the qualifying event as compared with patients with stroke or TIA. A similar pattern was observed in a large cohort from the Netherlands [6] and may indicate that cardiac events were perceived as more serious by the patients or, perhaps, more clearly linked to smoking.

Of note, the incidence of cardiovascular events and death was numerically higher in the group that continued to smoke after the index event compared with the NAILED population as a whole [37], despite a considerably lower mean age. This emphasizes the importance of finding effective methods to support smoking cessation in this very high-risk group. Provision of smoking cessation programs is listed as one of several interventions included in the WHO framework convention on tobacco control [46] that entered into force in 2005 and has since been ratified by 183 parties including Sweden. In accordance with the convention, progressive policy and legislative changes have been implemented during the last 20 years [47,48], and many countries have seen a concurrent decline in smoking prevalence in the general population [49]. However, the proportions of smoking cessation after MI and stroke have not changed [13–17], indicating failure when it comes to implementation of effective smoking cessation interventions for this group within health care. The 1-year proportions of smoking cessation in our study are similar to the corresponding proportions reported in Swedish quality-of-health-care registers [14,15]. In addition, our study contributes data on smoking status during long-term follow-up, showing that this proportion unfortunately also remained largely the same beyond 1 year.

Within publicly funded health care, as in Sweden, it is a constant challenge to use available resources in a way that maximizes health benefits for as many as possible. In that context, the design of the NAILED intervention, a pragmatic attempt to systematically implement recommendations from secondary preventive guidelines through a relatively low-resource-intensity approach, makes a valuable contribution. Whereas communication and education about the role of smoking should be integrated into standard care, more resource-demanding interventions directed toward smoking need to be carefully worked out to be feasible and effective among those most in need of support. The epidemiological results of our study provide increased knowledge about the pattern of smoking cessation after cardiovascular events and the factors associated with continued smoking, which is important to help identify those in need of support to stop smoking. According to our results, patients that do not quit in association with a cardiovascular event are unlikely to do so during long-term follow-up. It is also apparent that repeated advice as part of an annual follow-up of other risk factors is not sufficient to make a difference in this very high-risk subgroup of patients. Instead, future studies should focus on improving the evidence for the feasibility and effectiveness of more intensive smoking cessation support among patients that present at hospital with a recurrent cardiovascular event or patients that report continued smoking at short-term follow-up after a first-time event. Based on previous studies, such an intervention should consist of repeated counseling with motivational and behavioral elements for at least 1 month and, preferably, also an offer of pharmacotherapy with systematic follow-up as part of the intervention. To be clinically

relevant, it is important that studies include consecutive patients, carefully note the proportions of patients that decline participation or discontinue intervention, and assess both short-term and long-term effects on smoking cessation. To motivate both patients and the direction of resources to this type of intervention, it is important to increase knowledge of the quantitative effects of smoking cessation on new cardiovascular events and survival. Future studies should therefore also include these outcomes when evaluating the effects of smoking cessation interventions.

The current study has strengths as well as limitations. As part of the NAILED study, all consecutive patients in a geographically defined part of the country were screened for inclusion, resulting in a study sample with high resemblance of the background population in which multiple risk factor intervention for the prevention of recurrent cardiovascular events is indicated. The subgroup of current smokers in the NAILED study was not selected based on motivation to quit, and the outcome was analyzed according to the intention-to-treat principle, thereby providing realistic estimates of the effect of the intervention in clinical practice as well as epidemiological data with high external validity.

We did not use any objective method to verify the accuracy of self-reported smoking cessation, and thus it is possible that the proportions are slightly overestimated. We do, however, not find any reason to believe that the accuracy should differ between the treatment groups. In addition, the annual assessment of smoking status represented the point prevalence at those occasions; that is, we did not capture all smoking cessation attempts and relapses between contacts. Thus, the proportions of patients with registered attempts and relapses are probably underestimated. Also, the dichotomous assessment of smoking status did not capture quantitative changes in smoking habits, and therefore we do not know if there were any such changes over time or a difference between allocated treatment groups. Finally, we do not know to what extent smoking cessation support was provided in primary health care, what kind of support that was given, and to what extend pharmacotherapy was used in any of the groups. That limits our ability to analyze the overall trend toward a benefit for the control group regarding smoking cessation. Since the smoking cessation proportion in the control group was similar to corresponding data from Swedish quality registers, we do, however, believe that the support given was probably comparable to that in other regions of the country. It should also be noted that we did not adjust for multiple comparisons, increasing the risk that the between-group difference seen at 3 years could represent a type I error.

## Conclusion

Patients that do not quit smoking in association with a cardiovascular event are unlikely to do so during long-term follow-up, and continued smoking is associated with an increased risk of all-cause mortality. Annual, brief information and advice regarding cessation delivered by a nurse as part of long-term, multiple risk factor follow-up is not sufficient to improve the proportion of patients that quit smoking after ACS, stroke, or TIA.

## Supporting information

**S1 Checklist. CONSORT 2010 checklist of information to include when reporting a randomised trial\*.**
(DOC)

**S1 Table. Population characteristics associated with smoking status at 1 month and at the last follow-up occasion.**
(DOCX)

**S1 File. Definition of cardiovascular outcome events.**
(DOCX)

**S2 File. Study protocol.**
(DOC)

## Acknowledgments

The authors would like to acknowledge statistician Lars Söderström for his statistical guidance and for contributing to the statistical methods and integrity of the results.

## Author Contributions

**Conceptualization:** Anna-Lotta Irewall, Thomas Mooe.

**Formal analysis:** Anna-Lotta Irewall, Lina Åslund.

**Funding acquisition:** Thomas Mooe.

**Investigation:** Anna-Lotta Irewall, Lina Åslund, Thomas Mooe.

**Methodology:** Anna-Lotta Irewall, Lina Åslund, Thomas Mooe.

**Project administration:** Thomas Mooe.

**Resources:** Thomas Mooe.

**Supervision:** Anna-Lotta Irewall, Thomas Mooe.

**Writing – original draft:** Lina Åslund.

**Writing – review & editing:** Anna-Lotta Irewall, Joachim Ögren, Thomas Mooe.

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
