## [Decision Letter · Decision Letter 0]

29 Feb 2024

PONE-D-24-05204Smoking cessation and prognosis during long-term follow-up after stroke, TIA, and acute coronary syndrome—Results from the randomized controlled NAILED trialPLOS ONE

Dear Dr. Anna-Lotta Irewall,

Please address comments of reviewers.

Thank you for submitting your manuscript to PLOS ONE. After careful consideration, we feel that it has merit but does not fully meet PLOS ONE’s publication criteria as it currently stands. Therefore, we invite you to submit a revised version of the manuscript that addresses the points raised during the review process.

We look forward to receiving your revised manuscript.

Kind regards,

Hean Teik Ong

Academic Editor

PLOS ONE

Journal Requirements:

2. We note that you have selected “Clinical Trial” as your article type. PLOS ONE requires that all clinical trials are registered in an appropriate registry (the WHO list of approved registries is at      https://www.who.int/clinical-trials-registry-platform/network/primary-registries" https://www.who.int/clinical-trials-registry-platform/network/primary-registries and more information on trial registration is at http://www.icmje.org/about-icmje/faqs/clinical-trials-registration/).

Please state the name of the registry and the registration number (e.g. ISRCTN or ClinicalTrials.gov) in the submission data and on the title page of your manuscript.

a) Please provide the complete date range for participant recruitment and follow-up in the methods section of your manuscript.

b) If you have not yet registered your trial in an appropriate registry, we now require you to do so and will need confirmation of the trial registry number before we can pass your paper to the next stage of review. Please include in the Methods section of your paper your reasons for not registering this study before enrolment of participants started. Please confirm that all related trials are registered by stating: “The authors confirm that all ongoing and related trials for this drug/intervention are registered”.

Please see http://journals.plos.org/plosone/s/submission-guidelines#loc-clinical-trials for our policies on clinical trials.

3. In this instance it seems there may be acceptable restrictions in place that prevent the public sharing of your minimal data. However, in line with our goal of ensuring long-term data availability to all interested researchers, PLOS’ Data Policy states that authors cannot be the sole named individuals responsible for ensuring data access (http://journals.plos.org/plosone/s/data-availability#loc-acceptable-data-sharing-methods).

Additional Editor Comments :

Please address comments of reviewers.

Reviewers' comments:

Reviewer's Responses to Questions

**Comments to the Author**

1. Is the manuscript technically sound, and do the data support the conclusions?

Reviewer #1: Yes

Reviewer #2: Partly

2. Has the statistical analysis been performed appropriately and rigorously? 

Reviewer #1: Yes

Reviewer #2: Yes

3. Have the authors made all data underlying the findings in their manuscript fully available?

Reviewer #1: Yes

Reviewer #2: Yes

4. Is the manuscript presented in an intelligible fashion and written in standard English?

Reviewer #1: Yes

Reviewer #2: Yes

5. Review Comments to the Author

Reviewer #1: A very well-written article as part of the NAILED trial. The study showed that smoking cessation advise by a nurse had no effect on smoking cessation after a cardiovascular event. Moreover, the likelihood of quitting smoking would be early after a CV event and unlikely to do so in long-term followup.

It would be interesting to look into the amount of cigarettes and duration of smoking prior to the CV event and compare those who quit with those who did not as perhaps those who smoke longer and higher amuonts may have more difficulty in quitting. However as this study is a subanalysis it may be difficult to determine.

Overall this paper is good as it would change the practice and explore other ways for patients to quit smoking after a CV event.

Reviewer #2: This is a substudy of NAILED series RCTs. And the authors want to show the nurse-led telephone intervention on the smoking cessation and the consequent results. The primary endpoint is a negative result.

1. It is very confused of the setting of clinical endpoint. The intervention method seems to be nurse-led telephone follow-up, and what happened to this intervention should be foucused. The primary endpoint is OK. But the secondary endpoint was the association between early smoking cessation and prognosis, which is a little far away from the design.

2. The study is less of novelty. Although the secondary endpoint is positive, it dose not give novel knolweges.

6. PLOS authors have the option to publish the peer review history of their article (what does this mean?). If published, this will include your full peer review and any attached files.

Reviewer #1: No

Reviewer #2: No

---

## [Author Response · Author response to Decision Letter 0]

26 Apr 2024

Dear Editor and Reviewers,

Thank you for your time and for valuable comments on the manuscript. Please find our point-by-point response below. Reviewer #2 expressed concern regarding one of the secondary outcomes: The association between early smoking cessation and prognosis. After careful consideration, including the comments by reviewer #1, we have chosen to keep this outcome in the manuscript. The outcome of concern is presented as exploratory and we believe that it serves a value in describing the high risk associated with continued smoking, which is of importance to motivate further studies of interventions to improve early smoking cessation rates. If the Editor does not share our opinion, we are, of course, open to further discussion including suggestions of improvements. 

The resubmitted version of the manuscript is revised to adhere to the journal style requirements and a statement regarding trial registration has been added. All changes to the manuscript are marked in the manuscript with tracked changes file. 

Best Regards

Anna-Lotta Irewall, on behalf of all the authors. 

Point-by-point response

1. We have now revised the manuscript, figures and tables to adhere to the PLOS ONE style requirements as specified in the templates. Please let us know if there are any remaining formatting issues that we have not addressed.

2. We have now added information regarding trial registration to the title page. Please note that the NAILED trial appears in the ISRCTN registry under three different registration numbers. This is further explained under the trial registration headline in the methods section (page 10, lines 221-228). In accordance with the comment above we have added a statement confirming that all ongoing and related trials are registered (page 10, lines 228-229). The time period for participant recruitment and follow-up is found in the methods section (page 6, lines 111-112, and page 9, line 183). 

3. Data requests may be sent to Umeå University through the e-mail address registrator@umu.se. This address is given in the submission form. In Sweden the universities are legally obligated to store research data for a minimum of 10 years to ensure future availability. 

We have reviewed reference list and it should be correct. 

Review comments

Reviewer #1 comments:

“A very well-written article as part of the NAILED trial. The study showed that smoking cessation advise by a nurse had no effect on smoking cessation after a cardiovascular event. Moreover, the likelihood of quitting smoking would be early after a CV event and unlikely to do so in long-term followup.

It would be interesting to look into the amount of cigarettes and duration of smoking prior to the CV event and compare those who quit with those who did not as perhaps those who smoke longer and higher amounts may have more difficulty in quitting. However as this study is a subanalysis it may be difficult to determine.

Overall this paper is good as it would change the practice and explore other ways for patients to quit smoking after a CV event.”

Response: 

Thank you for your positive review. We agree that it would indeed be interesting to look into patterns related to the number of cigarettes smoked and duration of smoking. Smoking “dose” and duration of smoking would have been of interest both for the primary and for the secondary, explorative, outcome. It would also have been relevant to assess if the intervention had any effect on the number of cigarettes smoked among those who did not quit. Unfortunately, data collection regarding smoking was limited to smoking status at baseline (current/former/never) and each follow-up occasion (yes/no). We were therefore unable to perform any of those analyses. This is a weakness of the study which is mentioned in the last part of the discussion (page 24, line 502-504). 

Reviewer #2 comments:

“This is a substudy of NAILED series RCTs. And the authors want to show the nurse-led telephone intervention on the smoking cessation and the consequent results. The primary endpoint is a negative result.

1. It is very confused of the setting of clinical endpoint. The intervention method seems to be nurse-led telephone follow-up, and what happened to this intervention should be focused. The primary endpoint is OK. But the secondary endpoint was the association between early smoking cessation and prognosis, which is a little far away from the design.

2. The study is less of novelty. Although the secondary endpoint is positive, it dose not give novel knowledge.”

Response:

We do agree that the intervention effect on smoking cessation should be the main focus of the manuscript as that was the aim of this randomized controlled sub study. In our opinion, that is well reflected by the primary outcome and also by secondary outcomes with focus on between-group differences in smoking status at different time-points during follow-up. Correspondingly, these outcomes are also the main focus of the results and the discussion. In addition, we chose to include the association between smoking cessation and prognosis as an explorative, secondary outcome because assessment of cardiovascular risk and the effect of smoking cessation is of importance to motivate interventions to improve smoking cessation rates. Brief counseling is probably the most common intervention used to support smoking cessation in clinical practice and sometimes also included in guidelines despite little evidence for the effectiveness of such interventions among patients with cardiovascular disease. In this context, the neutral, main result of our study makes an important contribution and also the pattern of smoking cessation during long-term follow-up provide novelty. When it comes to the exploratory secondary outcome, we do agree that this result represents less of novelty as corresponding associations have been previously shown. However, reproduction of results in different populations does still hold a value and a major strength of this study is the populations’ high resemblance with the target population found in clinical practice.

---

## [Decision Letter · Decision Letter 1]

20 May 2024

PONE-D-24-05204R1Smoking cessation and prognosis during long-term follow-up after stroke, TIA, and acute coronary syndrome—Results from the randomized controlled NAILED trialPLOS ONE

Dear Dr. Irewall,

The revision submitted, R1, has satisfied matters raised by the 2 academic reviewers.  However, the article has been sent to a statistical reviewer, reviewer 3, on instructions of the administrative editor.  Several statistical points have to be addressed.  Please reply to the important comments raised by the statistical reviewer.

Thank you for submitting your manuscript to PLOS ONE. After careful consideration, we feel that it has merit but does not fully meet PLOS ONE’s publication criteria as it currently stands. Therefore, we invite you to submit a revised version of the manuscript that addresses the points raised during the review process.

We look forward to receiving your revised manuscript.

Kind regards,

Hean Teik Ong

Academic Editor

PLOS ONE

Journal Requirements:

Additional Editor Comments:

Reviewer comments adequately addressed.

Reviewer 3 comments:

Flowchart: If smoking status is checked after randomization, it is very unlikely to have same numbers of smokers the same between the intervention and control groups. There was no loss to follow-up over 8 years?

The stratification is unclear regarding the use of "either one factor or another". Is the randomization stratified by both factors? The stratification factor (e.g., Rankin scale) should be included in Table 1. Stratification factors need to show balanced in Table 1.

Power Calculation: Clarify that a chi-square test was used. Justify the effect size of a 20% difference. Additionally, if the sample size was not determined by power calculation during the study design stage, the post hoc power calculation is not meaningful.

If an intent-to-treat analysis was used, the analyzed sample size should be the same as the randomized number.

What model was used to handle competing risks?

For secondary endpoints, p-values should be adjusted for multiple comparisons, e.g., the p-value for comparisons at 3 months.

---

## [Author Response · Author response to Decision Letter 1]

22 Jul 2024

“Please review your reference list to ensure that it is complete and correct. If you have cited papers that have been retracted, please include the rationale for doing so in the manuscript text, or remove these references and replace them with relevant current references. Any changes to the reference list should be mentioned in the rebuttal letter that accompanies your revised manuscript. If you need to cite a retracted article, indicate the article’s retracted status in the References list and also include a citation and full reference for the retraction notice.”

In this second review of the reference list, we noticed that the article by Critchley et al had been retracted. In addition, we identified a very recently published update of the review by Rigotti et al. We have now replaced the retracted article by a more recently published Cochrane review by Wu et al (ref 5) and the review by Rigotti et al has been replaced by the update by Streck et al (ref 24). These changes to the reference list could not be done while using the “tracked changes tool”. Instead we have manually highlighted the new references in yellow. Please let us know if you find the update unclear in any way. 

Review comments:

Reviewer #3 comments:

“Flowchart: If smoking status is checked after randomization, it is very unlikely to have same numbers of smokers the same between the intervention and control groups. There was no loss to follow-up over 8 years?”

Baseline characteristics including smoking status were collected in-hospital for all identified patients prior to randomization. To clarify this, we have revised the first sentences under the sub heading Data collection in the methods section (pages 8, lines 169-172). With a sample of the current size (n=1890) the resulting groups are expected to end up well balanced in terms of baseline characteristics, all though some differences can still occur by chance. To minimize the risk of unbalance in certain key variables that could severely compromise the internal validity of the study, the randomization was stratified (further discussed below). The randomization was, however, not stratified for smoking status. Luckily for the current sub study, this variable did not deviate from the expected equal distribution between the randomized groups. 

Thank you for your valuable comment on the flowchart. Most participants did not take active part in the study assessments for 8 years. Depending on the year of inclusion, the maximum duration of follow-up was 3 to 8 years (described on page 8, lines 173-174, and in the figure legend). There was also participants that withdrew during the course of the study and participants in which follow-up was discontinued due to death. In response to your comment, we have now revised the flowchart (figure 1) to clarify discontinuation due to withdrawal or death. 

“The stratification is unclear regarding the use of "either one factor or another". Is the randomization stratified by both factors? The stratification factor (e.g., Rankin scale) should be included in Table 1. Stratification factors need to show balanced in Table 1.”

Thank you for pointing out ambiguity in the sentence describing the randomization process (page 6, lines 121-124). We have now rephrased the sentence by exchanging the word “subgroup” for “qualifying event”. We hope that you will find this change as clarifying. As requested, we have also added modified Rankin Scale as a variable in table 1. 

“Power Calculation: Clarify that a chi-square test was used. Justify the effect size of a 20% difference. Additionally, if the sample size was not determined by power calculation during the study design stage, the post hoc power calculation is not meaningful.”

We believe that an effect size of 20% is what could be considered a clinically relevant difference for this kind of intervention and we included a post-hoc power calculation to show that our study sample was large enough to detect such an effect with reasonable power. We do, however, acknowledge that the value of post hoc power calculations is a matter of debate and, after consideration, we have now removed this calculation from the manuscript as a response to your comment.

“If an intent-to-treat analysis was used, the analyzed sample size should be the same as the randomized number.”

The comparison of the proportion of non-smokers at 1 and 3 years did not include participants that had died by these respective time points, which we find reasonable. With that exception, the analyses did comprise all randomized participants included in this sub study (n=321). Death or withdrawal before the first study follow-up (1 month post discharge) was an exclusion criteria of this sub study and that could, perhaps, be seen as a violation to the intention-to-treat principle. However, the construction of the outcomes required smoking status to be registered at least once after hospital discharge and for that reason participants that did not remain in that study at 1 month could not be included in any analyses. As shown in figure 1, deaths and withdrawals between hospital discharge and the 1 month follow-up were well balanced between the study groups (n=11). 

“What model was used to handle competing risks?”

The competing risk analysis was performed according to Fine-Gray. This information has now been added to the statistical methods section (page 10, line 214). 

“For secondary endpoints, p-values should be adjusted for multiple comparisons, e.g., the p-value for comparisons at 3 months.”

We acknowledge the risk of type I error associated with multiple comparisons. However, several of the secondary outcomes were related (smoking status at one time point was highly predictive of smoking status at a later time point) and for that reason we believe that correction for multiple comparisons would have been too conservative in this study. Also, adjustment for multiple comparisons was not specified in the study protocol. After consideration, we have therefor chosen to keep the analyses for secondary outcomes without adjustment. Instead, we have now added the issue with multiple comparisons as a weakness in the discussion (page 24, line 491-492).

---

## [Decision Letter · Decision Letter 2]

29 Sep 2024

Smoking cessation and prognosis during long-term follow-up after stroke, TIA, and acute coronary syndrome—Results from the randomized controlled NAILED trial

PONE-D-24-05204R2

Dear Dr. Irewall,

We’re pleased to inform you that your manuscript has been judged scientifically suitable for publication and will be formally accepted for publication once it meets all outstanding technical requirements.

Kind regards,

Laura Kelly, PhD

Division Editor

PLOS ONE

Additional Editor Comments (optional):

Reviewers' comments:

Reviewer's Responses to Questions

**Comments to the Author**

1. If the authors have adequately addressed your comments raised in a previous round of review and you feel that this manuscript is now acceptable for publication, you may indicate that here to bypass the “Comments to the Author” section, enter your conflict of interest statement in the “Confidential to Editor” section, and submit your "Accept" recommendation.

Reviewer #3: All comments have been addressed

2. Is the manuscript technically sound, and do the data support the conclusions?

Reviewer #3: (No Response)

3. Has the statistical analysis been performed appropriately and rigorously? 

Reviewer #3: (No Response)

4. Have the authors made all data underlying the findings in their manuscript fully available?

Reviewer #3: (No Response)

5. Is the manuscript presented in an intelligible fashion and written in standard English?

Reviewer #3: (No Response)

6. Review Comments to the Author

Reviewer #3: All my concerns are addressed.

The statistics are acceptable.

7. PLOS authors have the option to publish the peer review history of their article (what does this mean?). If published, this will include your full peer review and any attached files.

Reviewer #3: No

---

## [Editor Report · Acceptance letter]

31 Oct 2024

PONE-D-24-05204R2 

PLOS ONE

Dear Dr. Irewall, 

I'm pleased to inform you that your manuscript has been deemed suitable for publication in PLOS ONE. Congratulations! Your manuscript is now being handed over to our production team.

Kind regards, 

on behalf of

Dr. Laura Hannah Kelly 

Staff Editor

PLOS ONE